# Foam Glass Granule Usage in Tile Glue Mixtures That Use a Reduced Portland Cement Amount

**DOI:** 10.3390/ma16031269

**Published:** 2023-02-02

**Authors:** Ramune Zurauskiene, Lijana Navickiene

**Affiliations:** 1Faculty of Civil Engineering, Vilnius Gediminas Technical University, 10223 Vilnius, Lithuania; 2Faculty of Medicine, Kaunas University of Applied Sciences, 50468 Kaunas, Lithuania

**Keywords:** tile glue, foam glass granules, reduced cement amount, white portland cement

## Abstract

In the last few years, ceramic tiles and tiles from natural rock with higher measurements were used. A huge amount of tile glue is used for high-measurement tile gluing due to a special gluing technology, which is characterized by a thicker glue layer. Due to this, a higher and higher amount of tile glue is used up during decorating. Regular tile glue mixture uses up to about 50–60% cement (according to mixture mass). In carried-out experiments, a lower amount of cement was used in tile glue mixture production (30%). Additionally, 5%, 10%, 15%, 20% and 25% of sand was replaced with small foam glass granules. These granules are made from glass waste. By using foam glass granules, lighter tile glue mixtures were produced, while reducing the cement amount can lower energy usage and CO_2_ emissions into the atmosphere. The main properties of tile glue were investigated as follows: flow of mixture, density, compressive strength, bending strength, tensile-adhesive strength, slip and water absorption. The properties obtained during the research prove that newly produced tile glue mixtures fulfill all requirements given to these types of mixtures. A total of 25% of foam glass granule from filler mass can be used in tile glue production.

## 1. Introduction

At construction sites, dry building mixtures are widely used. As modern production technologies and tools quickly develop, dry mixtures keep becoming more complicated and multicomponent. New mixture compositions, fillers and assortments are created and expanded upon. Building mixtures keep obtaining new desired or specific properties, using various fillers and complex admixtures that improve one or more mixture parameters.

One of these dry building mixture types is mixture for gluing tiles, also known simply as tile glue. It is a thin-layer mixture applied onto the base with a special jagged trowel. For tile glue, the European standard LST EN 12004-2:2017 is applied. Ceramic and tiles from natural rock were not only used for bathroom and kitchen zone decoration, but were more often used in other interior rooms. Tiles are not only used in building interiors, but also can be glued in the building facade. Depending on glued tile size, a different trowel with smaller or bigger jagged teeth is picked. Trowel teeth size determines the tile layer and glue mixture usage in 1 m^2^.

Trowel teeth thickness has to conform to tile size [1]. When picking the trowel, these sizes can be used as a guide: up to 100–6 mm height teeth; 100–200—8 mm; 200–330—10 mm; ˃330–˃10 mm. The higher the trowel teeth size, the more glue is used up. Additionally, if the other tile size is profiled, a trowel with big teeth or a trowel dedicated for average-sized tiles can be used.

As tile glue usage rises, so does both cement, and of the most commonly used glue filler—sand—demand rises when producing tile glue. For these dry mixtures, high amounts of cement and sand are regularly used up. Usually, these mixtures use about 50% cement and 50% clean quartz sand based on dry mixture mass. By lowering the cement amount in these mixtures, the CO_2_ amount, which is released when producing cement clinker, could be significantly decreased. Additionally, if there was a way to replace quartz sand with materials made from waste as raw material it would be possible to save good quality raw materials. This could save clean quartz sand and it could be used for other usages, for example, production of glass.

Natural fillers are usually used in dry building material production. Artificial light fillers can be used also. Light fillers are often used when it is desirable for mixtures to have specific properties, for example, when the mixture weight would preferably be lighter. Light fillers are produced or obtained from natural raw materials (expanded clay, perlite, agloporite, vermiculite, slag, pumice stone), and they can also be produced from waste (polystyrene foam granules, foam glass granules, polyurethane, plastic granules, technology waste granules and other granulated non-organic or sintered polymeric materials). In glue mixtures, fillers can be used that have a particle size that is relatively low—≤0.5 mm.

Foam glass granules are produced in up to nine-ten fractions. These granules can be both very small (0.04–0.125 mm) [2] or very big (8–16 mm) [3]. Glass waste can be recycled, though not only to foam glass granules. By using 800–900 °C temperature [4], three types of thermal insulation products can be made from crushed and milled glass powder: foam glass thermal insulative tiles [5], foam glass chips and foam glass granules. Foam glass granules can be gray [6] or white-colored [3]. White color and a small size is obtained from granules by using bleaching production process method as well as using raw materials that do not have dyeing oxides. White granules are perfectly suited for decorative mixture production. Coarse-sized granules are used as chimney insert filler between heat resistant insert and construction block. Foam glass granules are resistant to high temperature and changes to them do not occur at up to 1000 °C temperature. This was determined by adding foam glass granules into a ceramic formation mixture and burning it in 1000–1050 °C temperature.

Foam glass consists of gaseous and solid phases. Air gaps are pores, separated from one another by a thin glass film. Foam glass volume is made up of less than 8–16% solid phase [7], thus its density is quite low. In some cases, foam glass granules can be produced not only using glass waste, but also other waste, which can be of organic origin [8,9]. Produced foam glass granules are of low density [2], low strength [10] and low thermal conductivity coefficient [11]. Foam glass granules are used in composite production, and these composite properties are also investigated as follows: light concrete [12,13], armored light concrete [14] or mortar properties [15]. Mixing cement paste with foam glass granules, they spread evenly onto a small filler, while mixture matrixes and lighter filler contact zones are dense, which ensures light filler and mixture matrix common goal [12]. White foam glass granules are used in white decorative mortar mixtures. These dry mixtures in construction sites are filled with water and mixed, thus preparing them for usage in the very same construction site. These mixtures have great sound insulating properties and are attributed to the thermo-insulating mortar group.

By producing dry mixtures, all manufacturers’ goal is to ensure mixtures are plastic and smooth during the job, with standard physical and mechanical properties, strong, low-deforming during operation and resistant to environmental conditions and frost. Dry mixtures are usually modified using fillers. Modified dry mixture composition parts are the following: binding material, filler, polymeric filler—mixture mass thickening filler, modifying admixtures—plasticity rising or water absorption lowering admixtures [16]. Lately, as dry mixture amounts keep increasing (when gluing high-measurement tiles) it would be expedient to try and lighten these dry mixture masses. By making dry glue mixtures lighter, it would become possible to pack them into bigger bags, while in construction sites where gluing jobs are performed, these mixture usages would be lower. It should also be noted that the more these mixtures use materials made from waste, the less clean raw materials would be used, which could be perfectly used in other production areas.

After examining the literature, it was determined that the most commonly used filler in light concrete and plaster mortar production is foam glass granules. Cement mixtures with average coarseness and high coarseness foam glass granules properties are investigated, as well as their composite resistance to sulfate corrosion [17,18]. Even though foam glass granule usage in cement composites is currently investigated by researchers and there are submitted research results [2,4,6,10,12], foam glass granule usage in dry glue mixtures is not investigated enough. There is no investigation of foam glass granule effect on tile glue properties. All these processes were explored and investigated in this work, including trying to prepare white tile glue mixtures dedicated to mosaic and high-dimension rock tile gluing jobs, as well as trying to obtain glue mixtures with a lowered Portland cement amount.

## 2. Materials and Methods 

### 2.1. Materials

For research, white Portland cement CEM I 52.5 R (Aalborg Danmark) was used, which was produced from 95–100% clinker and 0–5% calcium sulfate additive (CaSO_4_·H_2_O) (EN 197-1:2011). Properties of Portland cement are shown in Table 1.

This work used quartz sand, which had a fraction of 0/1 and conformed to standard LST EN 12620:2005+A1:2008 requirements. The properties of the sand are shown in Table 2 while the mineralogical composition is shown in Table 3.

Two fraction white-foam glass granules (Ltd. “Stikloporas” Druskininkai, Lithuania) were used, 0.1/0.25 and 0.25/0.5. Granules were produced from clear window glass shards by melting them at a high temperature (around 850 °C) in a rotary kiln. Glass shards are crushed into powder and mixed with foamers in drum mixers. Afterwards, formed granules are heated in rotary kilns, separating them with a separating medium—metakaolin. After heating various diameters, granules are obtained which conform to LST EN 13055-1:2003 standard requirements. Foam glass granule properties are shown in Table 4. A foam glass granule-enlarged image is shown in Figure 1, granule-inside image is shown in Figure 2, the microstructure of a granule-surface image is shown in Figure 3, while X-ray analysis results are shown in Figure 4. Foam glass granules are covered with a melted cover, which has a low amount of open pores (Figure 2). Such foam glass granules are covered with small metakaolin particles. These metakaolin particles have a positive influence on Portland cement hydration and hardening [19,20]. Inside foam glass, granules have many pores and gaps; between coarse pores smaller ones are spread out, and between those ones even smaller pores (Figure 3). In analyzing research results (Figure 4), it can be noted that granules form an amorphous material, from which small cristobalite peaks can be observed.

Additives were used as follows: lime, calcium formate, methyl cellulose and starch ether. The used lime (LLC “Nordkalk” Pargas, Finland) was slaked lime. They increase glue mixture plasticity and slump while also improving work parameters. According to the manufacturer recommendations, up to 2% should be added from the mixture mass.

Calcium formate (GmbH “Chemische Fabrik Kalk” Koln, Germany) accelerates Portland cement initial setting and hardening. Calcium formate raises flow of mixture even with a small amount of water present, lengthens mixture usage period, raises adhesion, accelerates mixture spillage, raises hardened mixture strength and lowers self-shrinkage deformation. This filler perfectly matched with other admixtures which have plasticizing properties. According to manufacturer recommendations, it is added up to 2% from mixture mass.

Methyl cellulose ether filler (“Henkel KgA”, Düsseldorf, Germany) is used. Manufacturer recommends using 0.4% of this filler amount from mixture mass. It is a white-colored powder. This filler is used as a mixture thickener and water-retentive material, ensuring guaranteed mixture adhesiveness with various mineral bases.

Starch ether that melts well in cool water is made from corn starch. This stabilizing admixture regulates marginal shear stress size and plastic mixture viscosity, and does not allow for water and mixture to separate into layers when the flow of mixture is too high. A small amount of this additive is enough to thicken the structure and viscosity, but adding a high amount to mixtures can start making them exfoliate. The manufacturer recommends a 0.01–0.3% admixture amount in mixtures from mixture mass. This work uses yellow-coloured guarana ether powder made from a manufacturer in Finland.

### 2.2. Mixture Composition and Sample Production

In the initial investigation, a tile glue-sample mixture composition was picked according to other, already carried-out investigation recommendations and conclusions. Most researchers recommend and indicate a cement amount of 30–50% in composition from mixture mass.

In carried-out research, samples with glue mixture samples were used, when for their production a small amount of CEM I 52.5 R cement was used—30%. In regular tile glue mixtures, which are sold in building product shopping centers, cement amount was always higher (about 50–60% [21]). In this case, the lowest described in literature binding material amount was used, to confirm whether researched composition glue will fulfill glue requirements brought up in standards. The cement amount in the mixture was lowered on purpose, since white tile glue was mixed, which in its composition uses higher class (52.5 N) cement instead of the usually-used one in production (42.5 N). This was performed since white mixtures or decorative mortars usually use white high-quality cement, with the regular strength class of 52.5 N.

In the experiment, an amount of sand filler was replaced with foam glass granules. Granule amount was raised by 5% at every step up to 25%, thus lowering the sand filler amount. In total, six mixture compositions were mixed and investigated, and their compositions is shown in Table 5. The first sample composition (marked GM0) was benchmark, in which there were no foam glass granules. 

While dosing materials according to mass, it can be noted that G0 and G25 mixture volumes differ due to the fact that the changed sand bulk density is 1600 kg/m^3^, while the foam glass granule mixture density is 370 kg/m^3^. Due to this, in replacing 25% of sand in the mixture into 25% of foam glass granules, the volume of mixture G25 rises by 6.4%. This increase in volume is likely the reason behind some property changes, together with other factors such as lower foam glass granule strength compared to sand filler and different granule surface complexion.

According to the tile mixture composition shown in the table, six sample batches were formed in the laboratory with different filler sand and foam glass granule amounts, with the granule amounts being increased up to 25% in place of accordingly lowering the sand amount from mixture mass. The cement and admixture amount in sample mixtures were not changed.

In the laboratory, six sample batches were mixed and produced. Water amount was chosen so that the plastic fitting for work-tile glue mixture would be obtained (flow of mixture according to LST EN 1015-3:2002/A2:2007 120–140 mm). In every mixture, there was water used at 22% according to the mass (water cement ratio W/C 0.7). Tile glue mixture samples were produced, and mixtures initially were mixed by hand afterwards with a mechanical mixer, according to LST EN 196-1:2016. After mixing, the prepared glue mixture was left for 5–10 min for maturing and then mixed for 15 s. After completing a fresh flow of mixture experiments, prism forms were prepared and brushed with oil, their dimensions at 40 mm × 40 mm × 160 mm. Mixtures were put into them. Mixtures in forms were compacted using a vibration table and held for 24 h. After a day of hardening, the samples were removed from the forms and submerged into 20 ± 2 °C temperature water for further 27-day hardening.

### 2.3. Experiment Methods

Flow of mixture was determined after mixing. Flow of mixture was determined according to LST EN 1015-3:2002/A2:2007.

Hardened-tile glue density was determined according to LST EN 1015-10:2002/A1:2007. Sample bending and compression strengths were determined according to LST EN 1015-11:2020. The prisms (40 mm × 40 mm × 160 mm) were initially tested by loading in three points to failure and then compressed by being placed between two bearing plates of 40 mm × 40 mm. The machine has two steel supporting rollers spaced 100 mm apart, and a third steel roller of the same length and diameter located centrally between the supporting rollers. The load was applied at a uniform rate of 50 N/s. The mortars were compressed at a uniform rate of 200 N/s. A hardened-glue water absorption coefficient due to capillary determination was performed according to LST EN 1015-18:2003.

Hardened-tile-glue tensile-adhesive strength was determined according to LST EN 12004-2:2017. Sample machine was adjusted to tensile strength force experiment (Figure 5). The machine pulled a pulling tile with a 250 ± 50 N/s load using adapted and bending strength and not creating part.

Tile glue was applied onto a concrete tile with a trowel. Ceramic tiles with 50 mm gaps between them were put onto the glue. The 5 min open time method was used. After 5 min, every ceramic tile was affected by 20 ± 0.05 N load for 30 s. After 27 days of holding in standard conditions, pulling tiles were glued and glue tensile adhesive strength was determined by applying 250 ± 50 N/s rising energy evenly. Separate tensile adhesive-strength was determined using this formula:A_s_ = L/A, N/mm^2^;(1)
where A_s_—separate tensile-adhesive strength, N/mm^2^;

L—total tension load in newtons, N;

A—glued area in square millimeters (2500 mm^2^).

Dry pressing ceramic tiles that fulfill LST EN 14411 B I group requirements were used to determine glue tensile-adhesion strength. They are of very low porosity, and their water absorption is ≤0.5% according to mass. They are unglazed, smooth, have a matte gluing surface and have (50 ± 1) mm × (50 ± 1) mm measurements. For the hardened-gluing-mixture tensile-adhesion strength experiment, used samples are shown in Figure 6. 

The slip test was performed according to LST EN 12004-2:2017. The tile slip test was performed on freshly applied tiles. Steel verification ruler was secured onto the upper part of the concrete tiles with clamps in such a way that the verification ruler’s lower side would be horizontal when the tile is set into a vertical position. A 25 mm wide cover line without any gaps was added at the bottom under the steel verification ruler, while the concrete tile was applied with a thin layer of prepared glue layer in a square using a trowel. Then, on the concrete tile surface, the thinner glue layer was applied in such a way that it slightly covered the lower cover line’s edge. Glue was applied at a vertical angle with a jagged trowel into the verification ruler.

The cover line was then removed, 25 mm spacers were pressed near the verification ruler and after two minutes the ceramic tile was added near the spacers as shown in Figure 7 while being affected by 50 ± 0.1 N mass load. With ± 0.1 mm precision in three points between verification ruler and ceramic tile, gaps were measured using a caliper. Mass and spacers after 30 ± 5 s were removed, and the tile instantly and thoroughly was moved into a vertical position. After 20 ± 2 min, the gap between three points was again measured in the same way as before. All types of glue were tested. The experiment was performed with three ceramic tiles. Data were measured in millimeters and the average value is shown.

Tile slip test on freshly applied surface tile glue was conducted according to determined standard LST EN 12004-2 methodology. The experiment was performed with dry pressing ceramic tiles that have water soaking ≤0.5% according to mass, were unglazed, smooth, had matte gluing surface and had (50 ± 1) mm × (50 ± 1) mm dimensions.

The tile vertical slip was determined to be not lower than ± 0.1 mm precision in three points between the verification ruler and ceramic tile’s caliper. The highest ceramic tile’s mass-caused slip is the difference between the two reading values as shown in Figure 8.

Hardened-glue long-term water-absorption determination was performed according to LST 1476.5:1997. Hardened-tile glue-mixture-sample long-term water absorption experiment was performed by submerging the samples into water and then measuring the continuously submerged sample mass. Samples were submerged into a bath filled with clean drinking water, and the water level above the samples was not higher than 2–3 cm. Water absorption was determined after 1 h, 5 h, 12 h, 24 h, 48 h, 72 h and 96 h until a stable sample mass was reached. A stable mass was reached when two weightings every 24 h results differed less than 0.1 %.

Tile glue sample total open porosity was determined according to LST 1576.1:1999. Firstly, total sample absorption was determined [22]. Dried samples were put into a desiccator and vacuumed 1.92–2.22 kPa (from −0.98 to −1.02 atm) in such conditions for 1 h. After vacuuming, the desiccator was filled with water and air was let in inside. Afterwards, it was vacuumed again. The cycle was repeated a few times for length of time 1 h. After vacuuming, the samples were removed and put into a bath with water and held for 23 h. Afterwards, the samples were weighted. Sample absorption and total open porosity *W_R_* were calculated as follows:(2)$$W_{R}=\frac{m_{0}}{V}\cdot \frac{m_{4}-m_{0}}{m_{0}}\cdot 100,\%$$
where *V*—sample volume, cm^3^;

*m*_0_—dry sample mass, g;

*m*_4_—saturated in vacuum sample mass in air, g.

Granular material mineralogical analysis was performed using an x-ray research method using diffractometer DRON-7. Used anode—Cu, filter—Ni, anode voltage—30 kV, anode current—12 mA, goniometer gaps (0.5; 1.0; 1.5) mm. For X-ray decryption, the ICDD database was used.

Macrostructure research was performed using optic microscope “Motic” (“Microscope World”, Carlsbad, CA, USA) (magnification up to 100 times), and characterized zones were photographed using digital camera “Pixera PVC 100C” (GmbH “AV Stumpfl”, Wallern, Austria), connected to a computer. The foam glass granule microstructure was investigated with a scanning electron microscope SEM EVO LS 25 (Zeiss, Oberkochen, Germany).

## 3. Results

The flow of mixture-determined results are shown in Figure 9. The highest flow of mixture was obtained in glue mixture without foam glass granule filler GM0, which reached 135 mm. Almost the same flow of mixture results was obtained with the second (5%-foam glass granule amount in mixture from filler mass) and third (10%) batch mixtures, which were 134 mm.

An insignificantly lower flow of mixture results was obtained from the fourth and fifth batch sample mixtures. The smallest flow of mixture results was obtained with the sixth sample, i.e., the one that has the highest 25%-foam glass granule amount −129 mm. All sample flows of mixture result differences with the benchmark mixture go up to 5%. Flow of mixture lowering can be explained by the sand being replaced with granules which are lighter. Taking their determined amount into mass, their volume is higher than of the same mass sand. However, the flow of mixture does not differ that strongly—only up to 5%—due to the granule specific surface being lower than sand and their soaking requiring a lower water amount.

By adding foam glass granules into tile glue and mixing and light granules evenly spread in the mixture, lowering the glue density thus makes the tile glue samples lighter. Hardened-glue density results are shown in Figure 10. Depending on the added-foam glass granule amount, tile glue sample densities change accordingly every step by 6–7% between batches. 

The lowest density belongs to the GM25 batch samples, with the highest used at the 25% granule amount, with the average density of 930 kg/m^3^. The highest difference is between benchmark and sixth batch samples (GM25), which comprises 28%, as in, the higher the foam glass granule amount, the lower the glue density.

After a 28-day hardening, sample bending and compression strength was determined. Strength averages are shown in Figure 11.

Highest bending and compression strength is of benchmark sample, while the lowest is of the sample with the highest foam glass granule amount GM25. Bending strength lowered by 3.10 MPa (GM0) to the lowest 2.56 MPa fifth composition, with 20% foam glass granule amount. This difference is the highest, composing a 17.3% difference with the benchmark sample composition. It was determined that in raising the foam glass granule amount, the glue composition bending strength lowers. The least-changed were third (GM10) batch samples, while the most-lowered batch samples were from the fifth (GM20) batch. Due to this, it can be noted that the light filler does not significantly lower tile glue-bending strength, and such composition usage in mixtures make them more plastic and allows the use of them on surfaces that can deform, for example, heated floors.

As seen from the data, sample compression strength depends on the added-foam glass granule amount. Benchmark sample strength reached 9.10 MPa. That is the highest result between all of the investigated samples. The lowest compression strength was the fifth batch samples, with 20% foam glass granule amount. The average compressive strength of these samples was 6.69 MPa, 26.5% lower than the benchmark sample compression strength. From Figure 11, it can be seen that in increasing foam glass granule amount, sample strength properties have the tendency to lower. Such results determine foam glass granule structure fragility. The strength lowered, except for sixth batch samples, whose strength was slightly higher. 

Concluding obtained results, it can be noted that all batch samples fulfill minimal compression and bending strength requirements when installing tile decoration dedicated for operation on walls and floor.

It can be observed from shown data (Figure 12) that the highest tensile-adhesive strength gained through the experiment was obtained with the benchmark mixture sample without the foam glass granule filler was equal to 1.29 MPa. This adhesive strength is 158% higher than the minimal required value, according to LST EN 12004-1 standard.

The lowest values were obtained from the fifth (MG20) and sixth (MG25) batch samples; in both, the average was equal to 0.73 MPa, which is an adequate adhesive strength. Such strength was 32% higher than the minimal required standard value (LST EN 12004-2). Results are shown in Figure 12.

All cohesive failure happened within the adhesive layer according to standard classification (CF-A). These failures were found in all samples. An example of such cohesive failure can be seen in Figure 13.

After the experiment, it was concluded that all batch samples fulfill minimum standard requirements for tensile-adhesive strength, and confirmed that correct sample-mixture tensile-adhesion strength raising and also plasticizing admixtures were picked.

GM5-GM25 glue mixtures can be used when gluing traditional ceramic glazed tiles or absorbing stone mass tiles, used in kitchens, halls or in sanitary units on stable surfaces, for example, plaster base [1]. In this case, it is not necessary to use expensive glue so that the glued cover would be durable and serve a long time. In this case, it suffices to C1 class (according to LST EN 12004) glue mixture, and to have regular adhesion with base indicators. Such glue could be mixed from GM5-GM25 mixtures. The usage of such glue is as follows: using a trowel and forming 4 mm layer thickness about 1.8 kg/m^2^; 6 mm—about 2.5 kg/m^2^; 8 mm—about 3.2 kg/m^2^; 10 mm—about 4.1 kg/m^2^.

As seen in the given results (Figure 14), after the experiments it was determined that the highest slip was obtained with the first batch (GM0) mixture. Such slip was the highest out of all samples; however, it was still 16% lower than the allowed maximum required value, according to the LST EN 12004-1 standard. 

The lowest values were obtained with sixth batches (using 25% granule amount) samples, whose value average was 0.19 mm—which is 62% lower than maximum required standard value, or 55% smaller than control, i.e., a benchmark batch sample slip. Concluding on slip results, it can be noted that mixture admixtures were picked—water spread, adhesive and flow of mixture increasing, as well as other mixture components. Such composition glue mixture can be used not only in floor-tile gluing, but also for tile gluing on walls.

Long-term water-absorption determination experiments were performed by submerging samples into water and measuring obtained values after 1 h, 2 h, 5 h, 12 h, 24 h and 2–17 days. After carrying out long-term absorption experiments with completely hardened and dried tile glue samples, it was determined that absorption in the long-term increases by increasing the amount of added-foam glass granules. Obtained results are shown in Figure 15.

The foam glass-granule filler amount has a considerable effect on long-term absorption results. From Figure 15, it can be seen that MG5 and MG10 batch sample absorption was the lowest from those mixtures, which have foam glass granules. In lowering the sample density, the sample absorption rises. The highest absorption was after 17 days—30.1% was obtained with the highest foam glass granule amount of MG25. In the short period, from second to twelfth hour of soaking time, MG15 and MG20 batch sample initial absorption was lower than other batches with lower granule amounts. Fastest and firstly, water fills coarse pores, and later, over a longer time, fills smaller pores and foam glass granules. This can be seen when comparing absorption from one-hour absorption and absorption after 17 days. With regards to long-term absorption, it can be noted that hardened-glue mixture water fills gradually, while constant absorption is reached after a long time period, in this case 17 days. The difference between the benchmark sample batch (MG0) and the sixth sample batch (MG25) absorption after an hour was 19%, and after 17 days this difference reached 70%.

When samples were saturated with a water-using vacuum, it can be noted that after vacuum all batch sample absorption became higher than the long-term sample absorption when submerged into water (Figure 16). Obtained values were higher by 193%. It can be explained by water entering open foam-glass-granule pores during the vacuum, while the open foam-glass-granule pore amount is different and depends on granule size [8]. During the vacuum, all open hardened-material pores were filled, which, when soaking in water, would have gradually been filled with water. 

According to water absorption after vacuum results, the total open porosity was calculated. From Figure 17, it can be seen that the sample batch with the lowest porosity was in first benchmark batch samples without foam glass granule additive. The highest porosity was obtained with GM20 batch samples, which used 20% granule amount. The sixth batch-obtained porosity was lower. The higher the open porosity value, the higher the chance that resistance to frost for this material will be higher [23]. In other research work, it was noted that materials produced using glass have great bonding properties with hardened cement rock, which is a high resistance to frost indicator [24]. In other research work, it was investigated that 50–60% total open porosity was enough for resistance to a rise in frost value [25].

One of the main parameters describing tile glue water absorption intensity is water absorption coefficient due to capillary. The results obtained after the experiment are presented in Figure 18.

It can be seen in the figure that all batch samples do not have a very high water-absorption intensity rise due to added-foam glass granule filler. Absorption coefficient rise is proportional to the added-foam glass granule amount. In the figure, all W_c_1 category ranges are marked according to the LST EN 998-1:2017 standard (that is, the average category), which divides mortar into three categories according to water absorption coefficient due to capillary.

Tile glue with light-foam glass-granule-fillers capillary-water-absorption results show that the capillary water absorption values are not high. Such absorption can be explained by the initial soaking stage, when porous cement matrix and filler-foam glass granule surfaces touch with water, and cement stone gaps on the surface are filled. Further water penetration into further-hardened mixture layers is made difficult by obstructions. Water movement through capillary connections, both in the matrix and foam glass granules, is limited by the inner air steam pressure. Inner air steam formation makes intensive water absorption due to capillary difficult.

## 4. Conclusions

After completing research, it was determined that samples with foam glass granule properties are directly related with sand substitute, foam glass granule and amount added to tile glue. Added property-improving admixture determines different cement matrix properties as well as macro and micro structures, while foam glass granule additive determines tile glue physical and mechanical properties and structure.

It was determined that even in using the smallest picked cement amount (30%) instead of the regular production amount of 50–60% (according to mass) in adhesive tile mixture, it was possible to obtain tile glue fulfilling the main tile glue standard LST EN 12004 requirements. It can be noted that for white tile glue it is preferable to use higher class cement—52.5 N instead 42.5 N class—while quite significantly lowering its amount by a whole 20%. The tile mixture obtained in this work fulfills all raised operational property requirements.In this work, it was determined that all tile glue with foam glass granules batch samples surpass minimal tensile strength requirement. The optimal composite material amount in tile glue allowed the obtaining of appropriate vertical tile slip results, which fulfill standard requirements.Foam glass granule usage in tile glue mixture allows the lightening of glue mixtures, lowers their bulk density and allows up to 25% of materials made from recycled waste to be used in glue mixtures.A total of 25% of foam glass granule amount from filler mass can be used in tile glue production. That is the optimal amount of filler that fits the main brought-up tile glue standard LST EN 12004-2:2017 requirements.After reviewing the results, it can be noted that the amount of cement in tile glue production can be lowered, thus lowering the emitted CO_2_ amounts formed during cement production. Even though this work used high class Portland cement (strength class 52.5) its amount is significantly reduced—20 %. Due to this, it can be concluded that emitted CO_2_ amount for this product production will be lower when compared with regular white tile glues.

## Figures and Tables

**Figure 1 materials-16-01269-f001:**
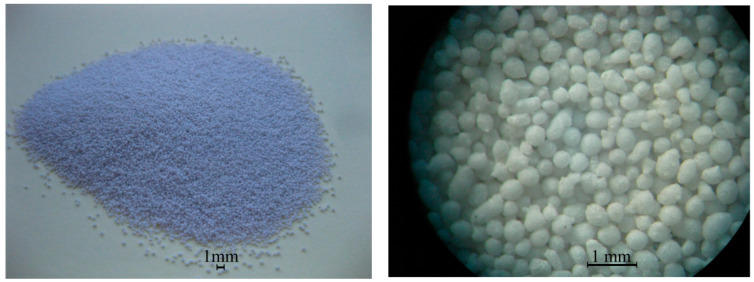
Foam glass granule (fraction 0.25/0.5) image.

**Figure 2 materials-16-01269-f002:**
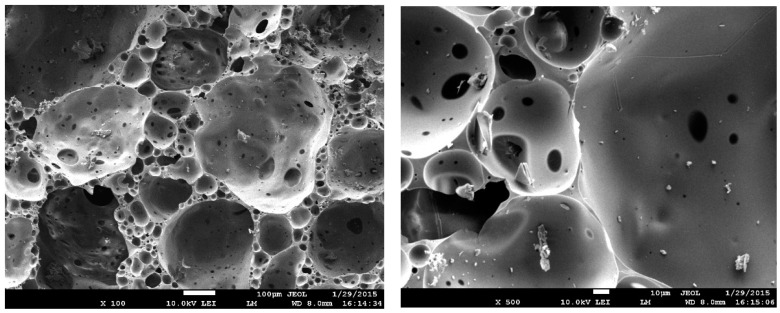
Foam glass granule inner structure microstructure analysis results: zoom in 100 times (**left**) and 500 times (**right**).

**Figure 3 materials-16-01269-f003:**
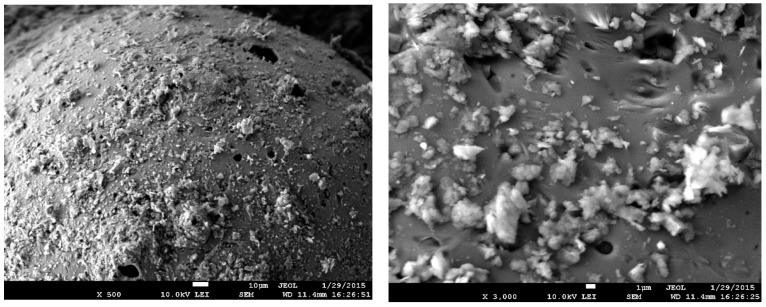
Foam glass granule surface microstructure analysis results: zoom in 500 times (**left**) and 3000 times (**right**).

**Figure 4 materials-16-01269-f004:**
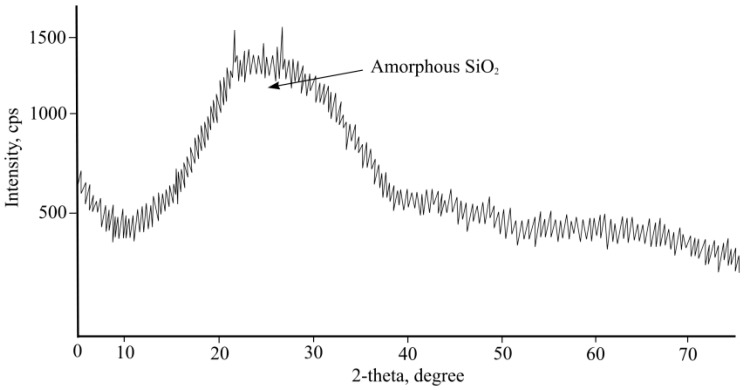
Foam glass granule X-ray analysis results.

**Figure 5 materials-16-01269-f005:**
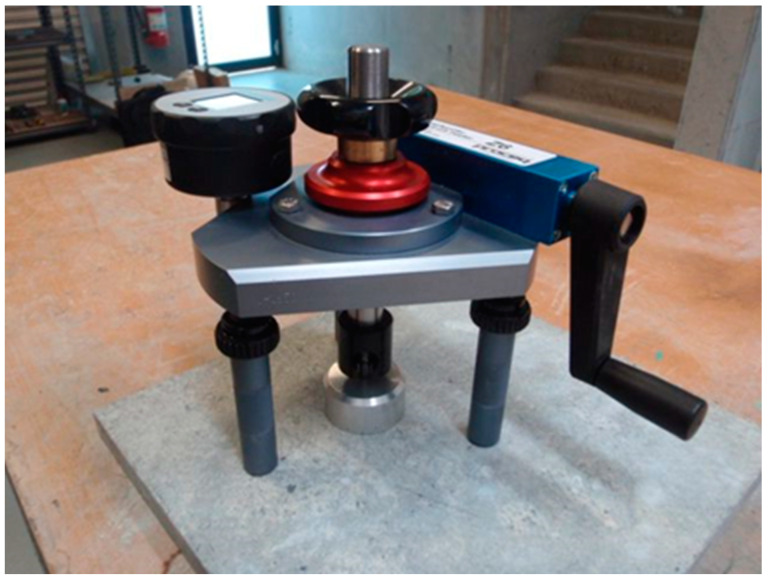
Mixture tensile strength measurement equipment.

**Figure 6 materials-16-01269-f006:**
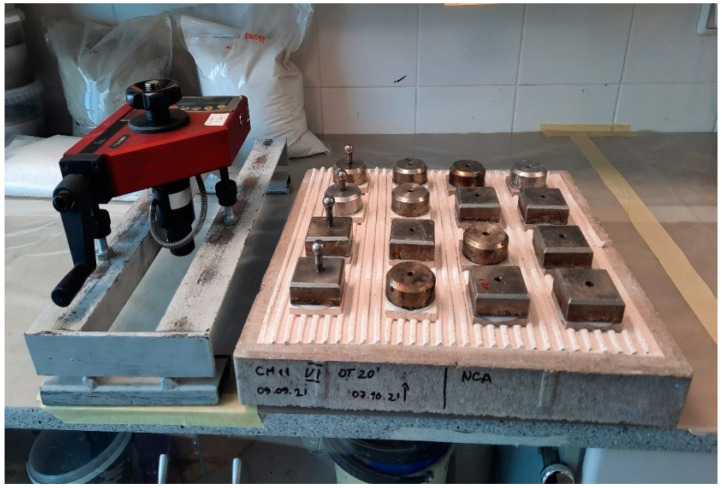
Samples prepared for tensile-adhesive strength experiments.

**Figure 7 materials-16-01269-f007:**
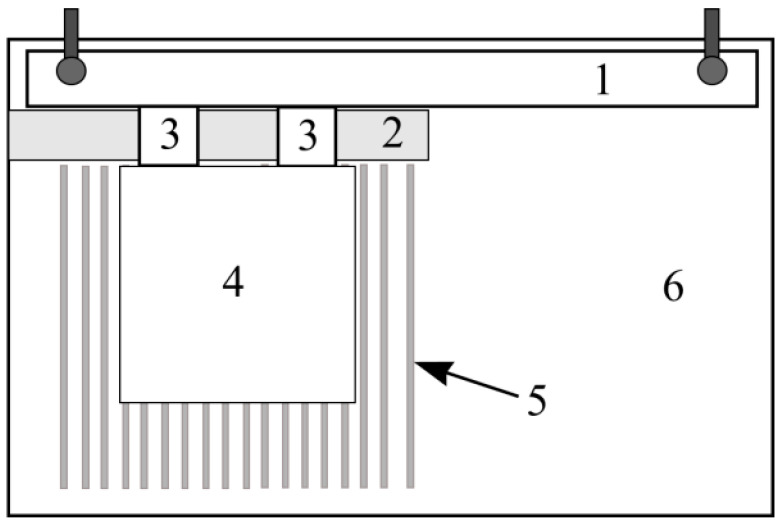
Slip experiment scheme: 1—steel verification ruler; 2—25 mm width cover line; 3—25 mm × 25 mm size 10 mm width spacers; 4—ceramic tile 100 mm × 100 mm; 5—glue; 6—concrete plate.

**Figure 8 materials-16-01269-f008:**
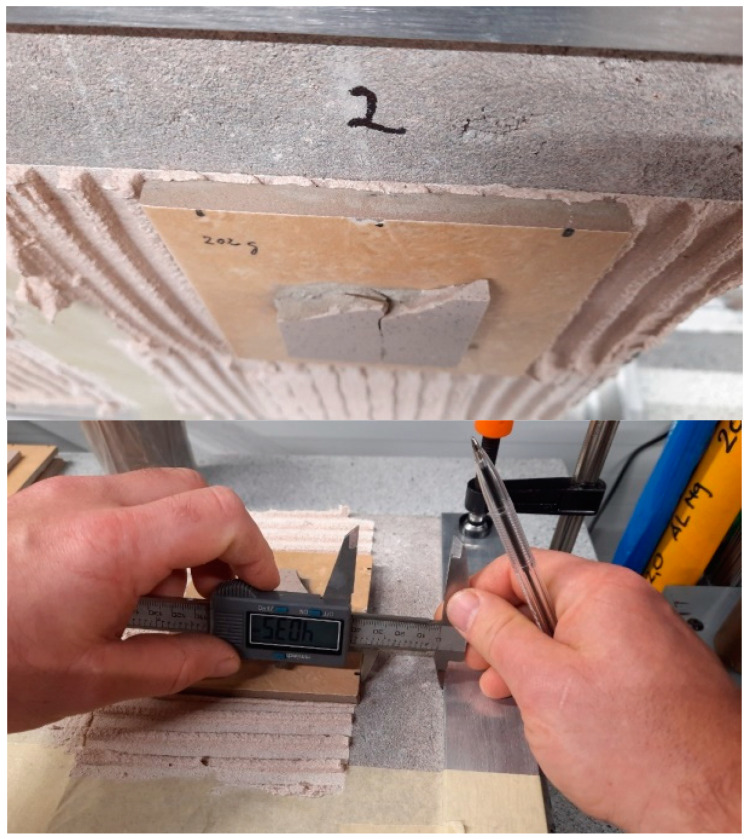
Slip test measuring process photo.

**Figure 9 materials-16-01269-f009:**
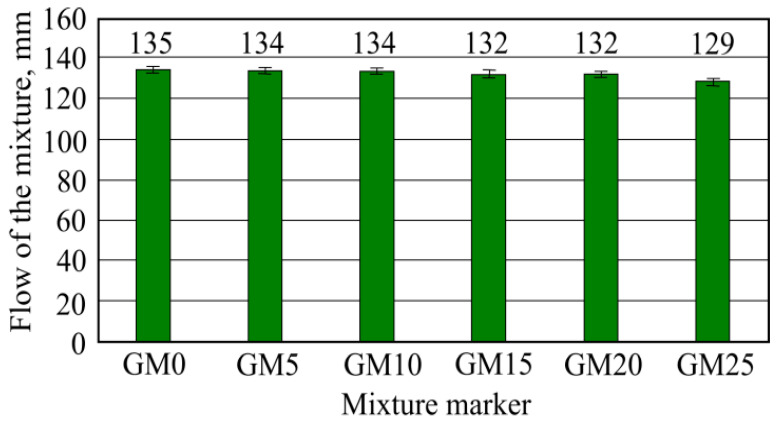
Flow of mixture results.

**Figure 10 materials-16-01269-f010:**
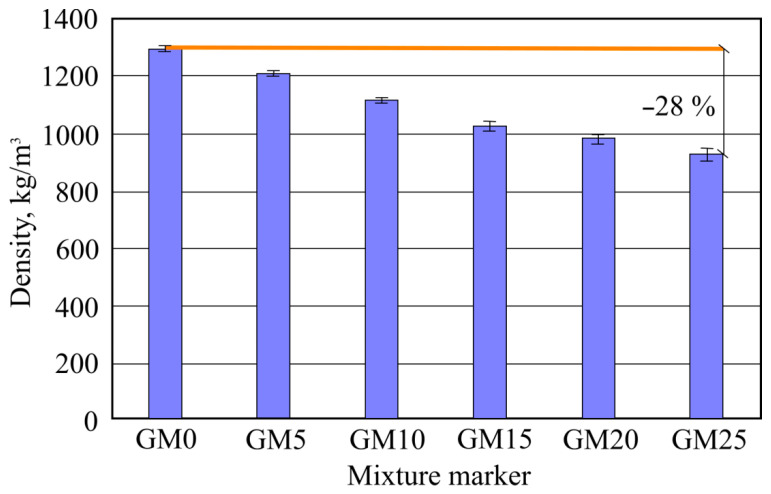
Hardened sample density results.

**Figure 11 materials-16-01269-f011:**
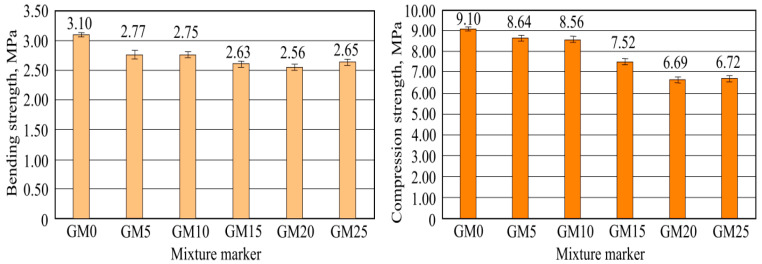
Hardened sample strength results: left—bending strength, right—compression strength.

**Figure 12 materials-16-01269-f012:**
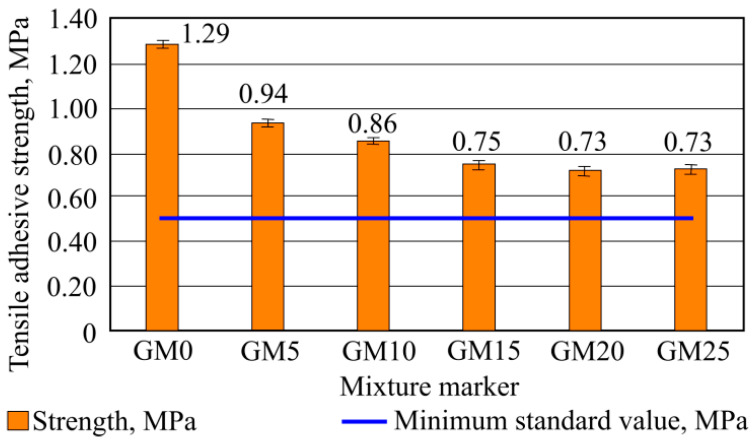
Tensile-adhesive strength experiment results.

**Figure 13 materials-16-01269-f013:**
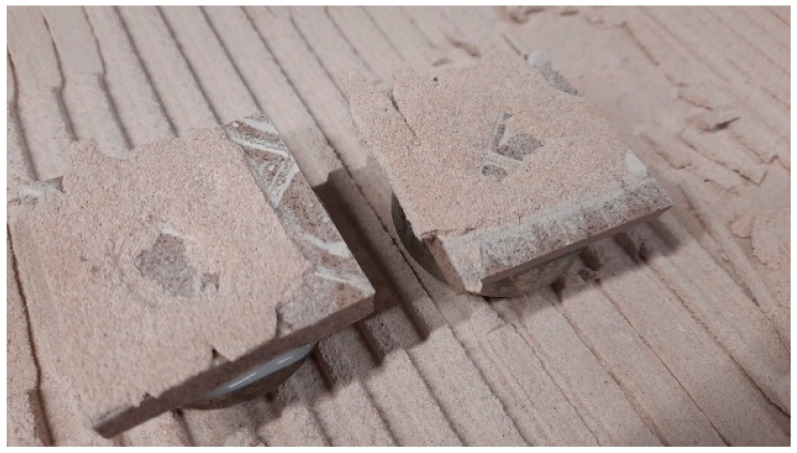
Photo of the samples after tensile-adhesive strength experiment.

**Figure 14 materials-16-01269-f014:**
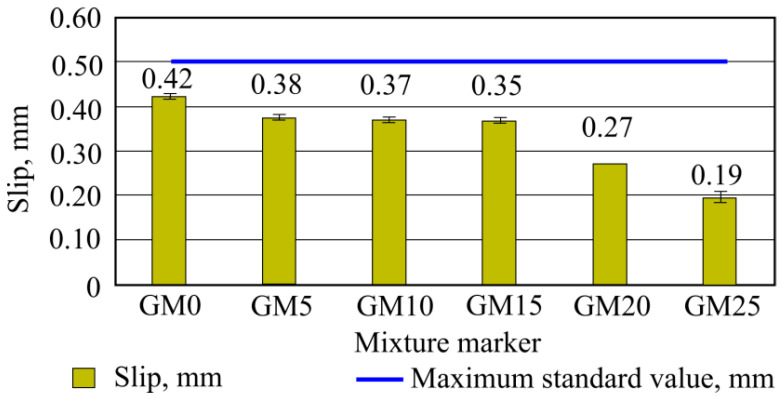
Slip test results.

**Figure 15 materials-16-01269-f015:**
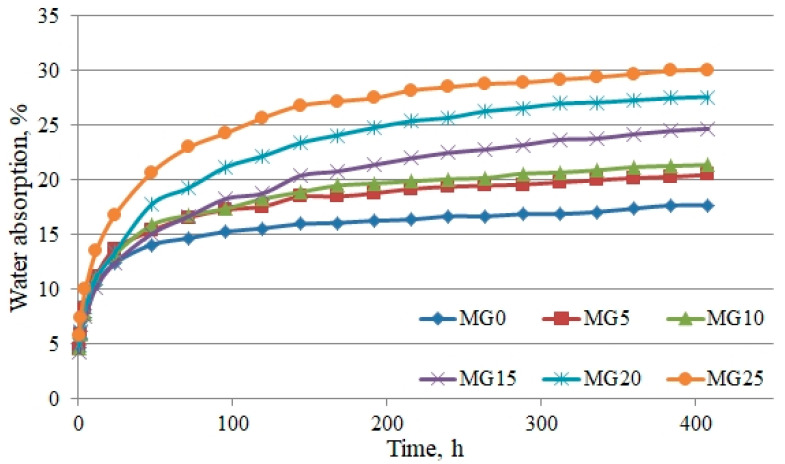
Water absorption kinetics.

**Figure 16 materials-16-01269-f016:**
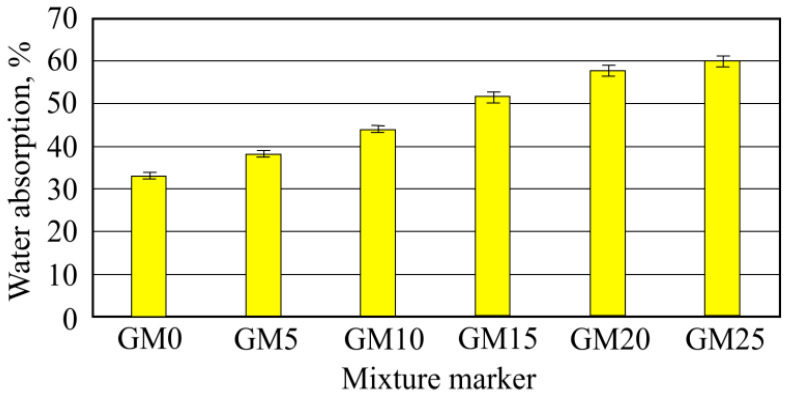
Water absorption results after sample vacuum saturation.

**Figure 17 materials-16-01269-f017:**
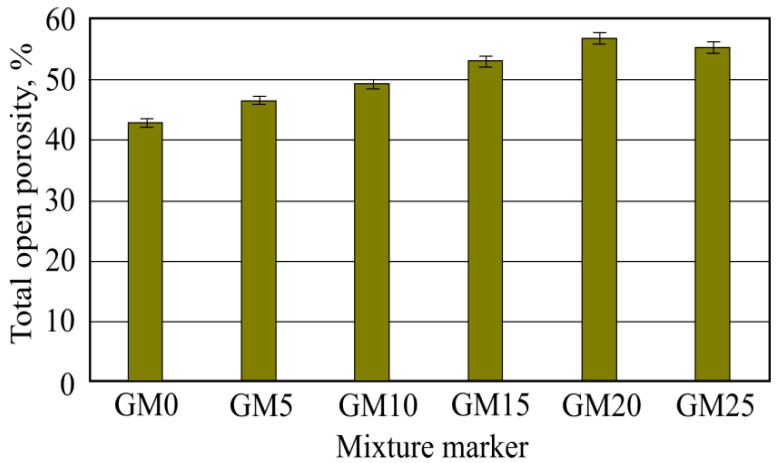
Total open porosity results.

**Figure 18 materials-16-01269-f018:**
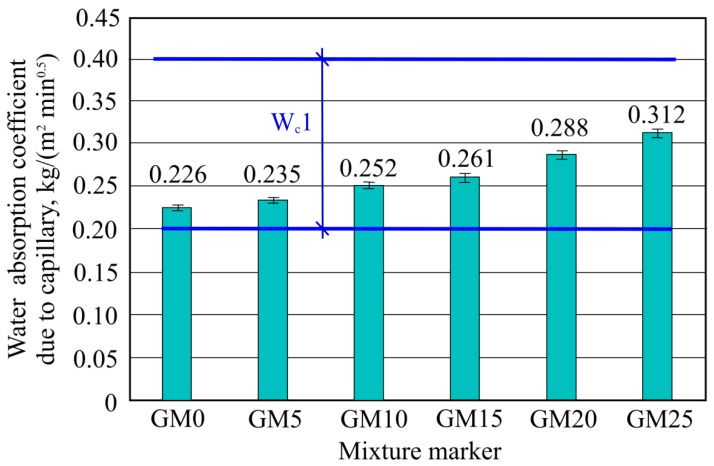
Water absorption coefficient due to capillary results.

**Table 1 materials-16-01269-t001:** Portland cement CEM I 52.5 properties.

Property	Value and Measurement Unit
Compression strength after 2 days of hardening	≥30 MPa
Compression strength after 28 days of hardening	≥52.5 MPa
Beginning of setting taking place not earlier than	45 min.
Soluble residue	≤5.0%
Glow loss	≤5.0%
Volume stability: expansion	≤10 mm
SO3 contents	≤3.5%
C3A	≤5.0%
Clorid contents	≤0.1%

**Table 2 materials-16-01269-t002:** Sand properties.

Property	Value and Measurement Unit
Particle density	2.65 g/cm^3^
Water absorption after 24-h soaking	˂1%
Water-soluble chlorides	≤0.01%
Sum of sulfur amount	≤1%
Sum of reactive rock amount	4.46%

**Table 3 materials-16-01269-t003:** Sand mineralogical composition.

Minerals	Amount, %
Quartz	62.8
Carbonates	18.0
Feldspar	15.6
Mineral aggregates (crystalline rocks)	1.8
Amphiboles	1.6
Mica	0.3

**Table 4 materials-16-01269-t004:** Foam glass granule properties.

Property	Granule Fraction	
	0.1/0.25	0.25/0.5
Bulk density, g/cm^3^	400	340
Thermal conductivity koeficient λ W/(m·K)	0.0798	0.0767
Resistance to crushing, MPa	1.57	1.51
Mass loss after 20 freeze-thawing cycles, %	1.1	1.2
Water absorption, %	10	15

**Table 5 materials-16-01269-t005:** Glue mixture composition and designation.

Sample Batch	Composite Mixture Parts *, % (According to Mass)							
	CEM	S	FGG (0.1/0.25)	FGG (0.25/0.5)	L	F	M	E
GM0	30	67.12	0	0	2	0.5	0.35	0.03
GM5	30	62.12	2.5	2.5	2	0.5	0.35	0.03
GM10	30	57.12	5	5	2	0.5	0.35	0.03
GM15	30	52.12	7.5	7.5	2	0.5	0.35	0.03
GM20	30	47.12	10	10	2	0.5	0.35	0.03
GM25	30	42.12	12.5	12.5	2	0.5	0.35	0.03

* CEM—cement, S—sand, FGG—foam glass granule, L—lime, F—calcium formate, M—methylcellulose, E—starch ether.

## Data Availability

The data presented in this study are available on request from the corresponding authors.

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
