# Peer review of "Foam Glass Granule Usage in Tile Glue Mixtures That Use a Reduced Portland Cement Amount"

_materials, 2023, doi:10.3390/ma16031269_

Round 1
Reviewer 1 Report
It is meaningful to use foam glass granules as the filler of tile glue and use solid waste. The research in this manuscript shows that foam glass granules used as filler in glue can achieve various properties required by the specification at a sand replacement rate of up to 25%, which is also a valuable result. However, readers are interested in why foam glass granules should be used in tile glue, and in addition to the use of solid waste, what technical problems can be solved by using foam glass granules should be explained clearly.
The water absorption of foam glass granules and quartz sand is very different. The test results show that the water absorption of the contrast sample MG0 and sample MG25 is 19% different in one hour absorption and 70% in 17 days absorption. However, the fluidity of the two has little change. What is the reason? The difference in specific surface area is explained in this paper, which is not deep enough. The specific surface area of the system is a factor that affects the fluidity, but the fluidity is mainly related to water, and the water absorption of the material is more important. Therefore, the test results need to be further clarified.
One of the point in this manuscript is that the use of foam glass granules can reduce the cement content, but it is not clear state in the paper. In the tile glue with a cement mass ratio of 30%, although foam glass particles are not used, and various properties of the glue are relatively good. It is suggested to analyze and compare the amount of cement in unit volume glue from the change of apparent density, and explain the reduction of cement.
In addition, the following aspects are suggested to be amended:
(1) Line 99, the material diameter range of foam glass particles is inconsistent with that in Table 5.
(2) The thermal conductivity in Table 4 is missing a unit.
(3) The positions of the two figures in Figure 2 and Figure 3 should be exchanged.
(4) In Figure 4, should there be the composition of kaolinite?
(5) Line 131: The redispersible latex written here is not used in the text.
(6) The 285 lines in the text indicate the specimen with the lowest bending and compressive strength, which is inconsistent with Figure 9.
(7) In the left figure of Figure 9, the bending strength value of GM10 seems abnormal, which is probably due to the test error.
Author Response
Thank you very much for the work you have done and I am providing the answers, we have taken into account all the comments
It is explained what problems are being tried to be solved using foam glass granules.
Fluency has changed little, the reasons are explained. The results have been clarified, the introduction has been supplemented, the bibliography has been supplemented, and the graphs have been supplemented. Cement reduction explained.
The first minutes when the mixture is mixed are important for spreading, at that time the particles of the mixture do not absorb water so strongly, as a result the spreading does not differ so much, and the effect of the additives is felt at the beginning.
Corrections made:
- Line 99, in Table 5. (corrected Table 4 and Table 5)
(2) Table 4 (fixed, units recorded)
(3) Figure 2 and Figure 3 (corrected, changed)
(4) In Figure 4, (we fixed the image title)
(5) Line 131: (we have corrected the description)
(6) The 285 Figure 9. (we reviewed it, corrected it)
(7) Figure 9, (we reviewed it, corrected it)
Reviewer 2 Report
The article concerns the study of the mechanical properties of the tile adhesive.
The subject matter is interesting and the scope of the conducted research and the results do not raise any objections. The article is written concisely, but requires corrections before publication.
1. The abstract should be corrected and supplemented with the scope of the conducted research.
2. The introduction should contain information about the problem or new solution and the expected results, but without details about the implementation. This is a scientific article, not a contractor's guide. What is the benefit of using lightweight aggregate?
3. There is no concise presentation of the standard requirements for tile adhesive in Europe/worldwide in the table.
4. Assumptions regarding the intended use of the adhesive (indoors or outdoors, type of substrate, vertical or horizontal substrate, special properties) should be given in advance and not in the summary.
5. Verify your keywords
6. The literature review should be updated.
7. Fig.1 Unify the graphics and add a scale. It's best to add new photos.
8. Use the layout of the IMRaD article. All research methods should be described in the Methods chapter, and all obtained results should be presented in the Results chapter. This part should be corrected.
9. The tests should be carried out in a wider scope: determination of transverse deformation, influence on high temperature (reaction to fire) ? Please provide information about these properties. If the mixture is to be used outdoors, the freeze/thaw resistance is also affected.
10. Tab. 5. It is better to specify the composition of the mixture in kg/volume.
11. Please complete the information on the testing equipment used, e.g. a testing machine.
Author Response
Thanks for the comments, corrections have been made on all issues.
- (Abstract is corrected and supplemented)
- (We added an introduction, provided information about the lightweight filler aggregate, and provided information about the problem and solution)
- (The requirements for tiles that are in the European Union are given separately for each property, so as not to resemble a manual)
- (Assumptions regarding the intended use of the adhesive we moved to another part)
- (Keywords to review and correct)
- (The literature review is updated, the literature on mortars and on research is presented)
- Fig. 1 (we added scale in the graphs and unified them)
- (We corrected the research methods, put all the research descriptions into one part according to the layout of the IMRaD article, we presented the results of the feature determination in a separate section)
- (Reaction to fire was not carried out, because the amount of organic matter in the mixture is less than 1 percent, therefore this characteristic, transverse deformations and resistance to frost are not determined, these characteristics are left for further research, the assumptions for the high resistance side were described and the indicators are presented in this article)
- Tab. 5. (the composition is given as a percentage of mass (kilograms)
- (We have added information about testing equipment and sample preparation)
Reviewer 3 Report
This is an informative and well written manuscript reporting on foam glass granule usage in tile glue mixtures that use a reduced Portland cement amount. After minor revision (see my specific comments), I would appreciate a publication of this article.
1. In this manuscript , high-grade cement is used to reduce the cement content, and it is suggested to add economic discussion.
2. The description of foam glass in the introduction should not be limited to the properties and characteristics, but also discuss its application and use advantages.
3. Line 92, what is the meaning of using sand ratio of 0/1?
4. Please add the components of the main materials.
5. What is the basis for the simultaneous change of the mass of sand and foam glass in the experimental mix proportion? Is it reasonable? How to determine the proportion of replacing sand with foam glass? And how to determine the water-cement ratio?
6. Lines 288-294 and 300-302. Are the main factors for strength reduction of these samples related to the reduction of sand content?
7. Lines 392-398, the porosity and frost resistance of the material should be subject to frost resistance test. It is recommended to refer to 10.1016/j.jclepro.2022.131527 to explain according to the test results.
8. Micro test should add to explain the micro mechanism of foam glass lifting ceramic tile adhesive.
Author Response
Thanks for the comments, corrections have been made on all issues.
- (We have included discussion items)
- (We described the foam glass granules, described their use in more detail, discussed their advantages)
- Line 92, (0/1 is the fraction of sand, which is written with the following abbreviation according to the filler Standard)
- (Components of the main raw materials have been revised and added)
- (Part of the sand was replaced with foam glass. This is based on the fact that we want to determine what percentage of foam glass we can put in place of sand. The ratio of water to cement was determined by mass.)
- Lines 288-294 and 300-302. (We have described and supplemented the reasons, and we have recorded this reason)
- Lines 392-398, (we explained based on specified to us 10.1016/j.jclepro.2022.131527)
- (Microstructural tests after disintegration were not performed at this stage, they were left for the second stage, other researchers have performed granule adhesion tests with hardened cement stone when studying concretes)
Reviewer 4 Report
It is an interesting topic and study, using foam glass granule to prepare tile glue mixtures, but the presentation of the data, of the results should be improved. For this reason, specific comments are provided as follows.
1. What is exactly the research necessity for tile glue mixtures field area? Please add in manuscript.
2. In the text, there is no corresponding result corresponding to line 80-82. Please revise.
3. In table 5, The quality of cement was constant at 30%, so how can it reflect the reduction of cement and its contribution to carbon emissions? In line 152, cement amount was 30–50 %, however, there was a lack of corresponding literature support. Please revise.
4. In the methodology and x-ray, some of the methods adopted for the study were not adequately described. Some important details were not provided e.g. sample preparation, sample number, etc. In addition, in the text, no corresponding test results were given, why?
5. For figs 8- 18, please add standard deviation bars.
6. The study in some sections is confusing. It would be great if the author can add graphical abstract or increase mechanism analysis.

Author Response
Thank you very much for the work you have done and I am providing the answers, we have taken into account all the comments.
The introduction was supplemented, the research results were adjusted, the literature was supplemented, and an additional description was provided.
- (Added area of application of tile adhesive, wider description, added description of components)
- (We reviewed and added the result)
- In table 5, (The chosen amount is lower than in the literature. Added literature.)
- (The description of the study and results has been corrected)
- For figs 8- 18 (Added standard deviation bars)
- ( The content of the chapters has been adjusted, the information of the chapters has been arranged sequentially)
Round 2
Reviewer 1 Report
The authors have responded to the modification suggestions put forward by the first review. The following two points need to be further modified and supplemented:
(1) The expression of Figures 2 and 3. The author misunderstood the modification opinion. The modification opinion does not mean that the positions of Figure 2 and Figure 3 should be exchanged, but that there are two pictures in Figure 2 and Figure 3, and the positions of these two pictures should be changed. Please note that the left and right positions in the figures are inconsistent with the marked ones.
(2)The use of foam glass particles to reduce cement consumption is still unclear stated. It is inappropriate to just add a sentence to the conclusion. It is still recommended to analyze and compare the amount of cement in unit volume of glue from the change of apparent density to explain the reduction of cement.
Author Response
Thank You very much for your comments, due to rushing last time I did not notice and did not understand Your comment about Figures 2 and 3. I have since corrected it. About the sentence in the conclusion, thank You very much for the observation. I have now written the conclusion so that it would fit with all research and article content and I have also added reflections about volume increase and influence on obtained properties due to this factor. I have also clarified why this cement was used. Thank You.
Reviewer 4 Report
I don't have any suggestions
Author Response
Thank You very much for Your work and comments.